# Physical Activity, Body Image, and Its Relationship with Academic Performance in Adolescents

**DOI:** 10.3390/healthcare11040602

**Published:** 2023-02-17

**Authors:** José Pedro Guimarães, Juan Pedro Fuentes-García, Jara González-Silva, María José Martínez-Patiño

**Affiliations:** 1Faculty of Psychology, Education and Sports, Lusofona University of Porto, 4000-098 Porto, Portugal; 2Faculty of Sports Sciences, University of Extremadura, 10003 Cáceres, Spain; 3Faculty of Education and Psychology, University of Extremadura, 06006 Badajoz, Spain; 4Faculty of Education and Sports Sciences, University of Vigo, 36005 Pontevedra, Spain

**Keywords:** physical activity, academic performance, body image, secondary school, adolescents

## Abstract

Academic success in adolescence is a strong predictor of well-being and health in adulthood. A healthy lifestyle and moderate/high levels of physical activity can influence academic performance. Therefore, we aimed to assess the relationship between the physical activity levels and body image and academic performance in public school adolescents. The sample consisted of 531 secondary school students in Porto (296 girls and 235 boys) aged between 15 and 20 years. The study variables and instruments were satisfaction with body image (The Body Image Rating Scale), assessment of physical activity (International Physical Activity Questionnaire for Adolescents (IPAQ-A), assessment of academic performance (academic achievement), school motivation (Academic Scale Motivation). The statistical analysis performed was descriptive analysis, an analysis of covariance, and a logistic regression. Regarding the results obtained, although there was no association between physical activity level and academic performance, it was observed in 10th grade students that the school average was higher for those practicing group or individual sports compared to students practicing artistic expression. Regarding the level of satisfaction with body image, we found different results in both genders. Our results support the importance of an active lifestyle, with the presence of regular physical activity being an important factor in improving academic performance.

## 1. Introduction

Academic success in childhood and adolescence is a strong predictor of well-being and health in adulthood [1]. Adolescence, for young people, is a period characterized by psychological, physical, and behavioral transitions that can influence their health and well-being in adulthood. The maximization of time spent on academic tasks has led to an increase in the extinction of extracurricular activities, thus decreasing the time spent on physical activity (PA) [2].

As mentioned by Nicollucci and Maffeis [3], physical inactivity is one of the main factors leading to childhood [4]. According to Guthold [5] in 2010, 81% of adolescents aged 11–17 years were considered insufficiently active. Lack of PA is one of the main risk factors for global mortality, affecting overall health worldwide [6]. Specifically, regular PA practice has a positive long-term impact on body composition, cholesterol, blood pressure, blood glucose, muscle strength, aerobic capacity, motor skills, and bone development [7]. However, it is important to emphasize that the benefits of PA are not limited to physical health, also influencing mental health [8,9,10].

Specifically, early PA recommendations state that adolescents should engage in PA every day as part of their lifestyle and that they should engage in at least 20 min of moderate to vigorous PA, three sessions per week [11]. Subsequently, Zhang et al. [12] suggested that children and adolescents should participate in at least 60 min of enjoyable PA every day at moderate to vigorous levels of exertion.

To design efficient strategies to increase PA and sport practice in children and adolescents, it is essential to know their preferences. In the literature, we can find several models that try to explain the factors that influence the practice of PA and distinguish the demographic and biological factors; psychological, emotional, or cognitive; behavioral; socio-cultural; environmental and the characteristics of PA [13].

Gender and age seem to be factors influencing PA practice. Zhang et al. [12] suggested that boys are more active than girls. Corroborating this, Mota and Sallis [14] stated that gender differences can be explained by different social influences: boys are more encouraged to practice PA, have more and different opportunities outside school, and reveal more positive PA experiences than girls. However, it is important to note that, in the last decade, progressive changes in the social role of women have contributed to an increase in the proportion of active women compared to men [15]. The same author states that PA decreases drastically with age, with the decrease being more noticeable in girls. While PA practice during childhood is a key predisposing factor for future practice, adolescents should be a target population for interventions that value PA in their free time, after school, and on weekends [14].

The psychological, emotional, and cognitive aspects are found within one of the four groups where the benefits of PA are framed, specifically within the group of mental and cognitive health benefits where, in addition to the aspects above-mentioned, body image and self-concept are also found.

Regardless of the gender differences, PA seems to play an important role in body image. Indeed, such a relationship has been studied, suggesting that PA is positively related to body image in both men and women [16]. In general, those who practice PA seem to experience a healthier body image (i.e., higher satisfaction or lower dissatisfaction with their body image) compared to those who do not practice PA on a regular basis.

The mechanisms that explain the effects of PA on body image perception are shown in a literature review developed by Martin-Ginis et al. [17]. According to the authors, the relationship between body image and PA can be explained by objective changes in physical fitness, changes in perceived physical fitness, and changes in self-efficacy. In addition, individual characteristics (such as age and gender) [18] and PA characteristics (such as type, intensity, and frequency) should be considered when analyzing the relationship between body image and PA [17].

On the other hand, regular PA can also lead to better academic performance [19]. Abalde-Amoedo and Pino-Juste [20] showed the close relationship between PA and physical and mental health, reflected in academic performance and improved cognitive processes. The same authors highlight the relationship between childhood obesity and overweight (which may be caused, according to the authors, by physical inactivity) with lower academic performance.

Most studies examining the association between PA and academic performance in young people used self-reported measures of physical activity. In general, these studies suggest a positive association between PA and academic performance [21]. However, we found studies that did not find this association in young people. Nelson and Gordon-Larsen [22] and Huang et al. [23] found a negative association between moderate to vigorous PA and academic performance.

The study of the influence of PA on academic achievement is currently urgent. Studies on this aspect could be used to support policy, school, and community interventions. Thus, the aim was to assess the relationship between the PA levels and body image and academic performance in secondary school adolescents.

## 2. Materials and Methods

### 2.1. Sample

The sample consisted of 531 secondary school students (grades 10, 11, 12) from secondary public schools from the Porto District (296 girls and 235 boys) aged 15–20 years (16.63 ± 1.10). Three classes were randomly assigned to the study by the school principal, one for each school year. Therefore, to participate in the study, it was necessary to be a secondary public-school student, and not all primary and university students could participate. The population was selected according to the availability of the public school to participate in the study. Forty-seven secondary schools in the Porto District were selected, taking into account the national ranking according to the averages of the national exams, highest, medium, and lowest graded. The schools were contacted by email to schedule a face-to-face meeting with the school principal, and consequently, to present the study. Nine schools responded and agreed to participate in the study (Figure 1).

All study participants were informed of the study objectives, and the parents or guardians of each participant provided their written informed consent for their child/student to participate. The study was approved by the Ethics Committee of the University of Vigo (Cod: 33323136T; CAPD: 30/09/2019); this was conducted in accordance with the Declaration of Helsinki for Human Studies of the World Medical Association.

### 2.2. Variables and Measuring Instruments

Anthropometric measurements: Height and body weight were self-reported. Body mass index (BMI) was calculated using the weight/height ratio^2^ (in kilograms per square meter). Subjects were classified according to the World Health Organization’s age-adjusted BMI z-score cut-off point [24] as normal weight (BMI z-score ≤ +1SD), overweight (+1SD < BMI z-score ≤ +2SD), or obese (BMI z-score > +2SD).

Assessment of habitual PA: The International Physical Activity Questionnaire for Adolescents (IPAQ-A) [25] validated for the Portuguese population [26] was used. This instrument was developed to assess the general PA levels of adolescents. It consists of nine items; the total score is derived from only eight of them on a 5-point Likert scale. The final score is given by the sum of the eight items ranging from eight points to 40 points, where lower scores indicate lower levels of PA. For this study, the participants were classified into sex-adjusted tertials of PA level as follows: low level of PA (girls: ≤1.76 points, boys: ≤2.21 points), medium level of PA (girls: 1.76–2.46 points, boys: 2.21–2.85 points), and high level of PA (girls: ≥2.46 points, boys: ≥2.85 points).

In addition, the participants were asked about the practice of extracurricular physical activity, and if so, the modality/ies practiced. The participants were classified into six groups according to the types of PA practiced: team sports (handball, football, basketball, volleyball, indoor football, and roller hockey), individual sports (swimming, athletics, cycling, gymnastics, badminton, and tennis), gym sports (Zumba, weight training, and fitness), martial arts/combat sports (judo, karate and taekwondo), artistic expression (ballet, dance, contemporary, hip-hop, skating and rancho), and multiple sports (i.e., including the practice of more than one modality).

For the classification of PA, the Evenson [27] cut-off points were considered in which the PA is classified according to the period of PA carried out in one day. Specifically, the categories were sedentary physical activity (0–100 min/day); light activity (101–2295 min/day); moderate activity (2296–4011 min/day); or vigorous activity (≥4012 min/day)

Satisfaction with body image: The Body Image Rating Scale of Gardner et al. [28] was used. This tool includes 24 sequenced figures, where 12 represent men and 12 women (Figure 2). The figures are ordered from the smallest (1) to the largest (12). Thus, each participant indicated, by circling the silhouette number, the one that represented the current perception of their body image and the silhouette number they would like to have. The discrepancy between these two responses was considered as an indicator of dissatisfaction with body image. Thus, for values equal to zero, the participant was satisfied with their body image. On the other hand, any value other than zero was considered dissatisfaction with body image.

Assessment of academic performance: The students’ self-reported final assessments from the previous year were used to assess academic performance. Students in 10th grade reported the final average obtained in 9th grade (scale from 1 to 5), students in 11th grade reported the average of 10th grade (1 to 20), and students in 12th grade reported the average of 11th grade (1 to 20).

Academic performance as an educational goal is usually evaluated using the evaluation scale used in the respective educational systems, that is, school grades (consulted in the school record or self-reported) [29]. In educational systems where school records are defined as letter scales (A, B, C, etc.), each letter is often assigned a number. Specifically, in Portugal, there are two rating scales, rating scale 1, which is an increasing numerical scale from 1 to 5 and is applied up to the 9th grade, and rating scale 2, an increasing numerical scale with values between 0 and 20, applied from the 10th grade.

School motivation: The Thornberry [30] scale was used to assess school motivation. The test consists of 18 items, composed of statements to assess school motivation using a Likert-scale and grouped into three groups.

Achievement motivation: This assesses the student behaviors that orient students toward achieving success in the tasks assessed with standards of excellence (items 3, 6, 8, 10, 11, 15, 17). 

Causal attributions of achievement: Assesses the student-generated explanations of the cause of their academic results (items 1, 4, 9, 12, 14, 18). 

Self-efficacy: This assesses the student’s perception of their own ability to successfully perform academic tasks (items 2, 5, 7, 13, 16). This scale uses a correction model for its classification: high scores reveal higher levels of academic motivation, while low scores reveal lower levels of the variable being assessed. Thus, most of the items are scored with the following scores: “I always think and act like this” is worth 2 points, “I sometimes think and act like this” is worth 1 point, and “I never think and act like this“ is worth 0 points. However, it should be noted that some of the items are written negatively (i.e., they describe behaviors or thoughts that are characteristic of a low level of motivation). Therefore, the scores of these items were reversed and, in the case of the adapted instrument, were: 1, 3, 5, 7, 9, 12, 15, 17.

### 2.3. Statistical Analysis

Statistical analysis was performed using BMI SPSS Statistics for Windows (Version 25.0. Armonk, NY, USA: BMI Corp.). Descriptive analysis of the variables was performed using the median and 25th and 75th percentiles for the continuous variables and absolute and relative frequencies for the nominal and ordinal variables. The Chi-square test was used to compare the proportions. The normality of the variables was studied using the Kolmogorov–Smirnov statistical test.

Analysis of covariance (ANCOVA) was used to compare the school mean according to the PA level tertials, school mean according to the out-of-school physical activity practice, school mean according to the type of out-of-school physical activity, and the school mean according to body image satisfaction. The mean values presented were adjusted for the following confounding factors: age, gender, parents’ best educational level, and, if applicable, school motivation, and level of physical activity. Homogeneity of variances was checked using Levine’s test. For multiple comparisons, the Bonferroni post-hoc test was used.

The association between body image satisfaction and PA level was tested using logistic regression models stratified by gender and adjusted for age, better parental education, and body mass index. Odds ratios (OR) and their 95% confidence intervals (CI) were used to express the magnitude of the associations.

A significance level was considered for a *p*-value < 0.05.

## 3. Results

### 3.1. Characterization of the Sample

Table 1 describes the characteristics of the total sample by gender. Most participants were normal weight (80.5%), 17.9% were overweight, and 1.5% obese, with no statistically significant differences between genders. In addition, 82.4% of the participants were found to be dissatisfied with their body image (girls: 80.2% vs. boys: 85.1%, *p* = 0.149).

Boys were found to be more physically active than girls (2.6 (2.0; 3.0) vs. 2.1 (1.7; 2.6), respectively, *p* < 0.001) and had a higher proportion in extracurricular PA (64.7% vs. 52%, respectively, *p* = 0.002). Regarding the type of extracurricular PA, there was a higher pro-portion of girls in individual sports (23.4%), academic (18.8%), and artistic expression (34.0%) compared to boys who showed a greater predisposition to team sports (46.7%). There was higher school motivation among girls compared to boys (26.0 (22.0; 30.0) vs. 25.0 (22.0; 28.0), respectively, *p* = 0.020).

No statistically significant differences were found between the genders for the variables: age, weight, height, average schooling by year of schooling, dissatisfaction with body image, and difference in perceived and desired body image.

### 3.2. Relationship of Physical Activity Level, Extracurricular Physical Activity, and Academic Performance

There were no statistically significant differences between academic performance and PA level by year of schooling (*p* > 0.05, for all) (Table 2).

Table 3 presents the academic performance, by year of schooling, according to the practice of extracurricular PA (yes/no) and the type of activity practiced. We found that for students in grade 10, the academic performance was higher for those practicing group or individual sports compared to students practicing artistic expression (4.7 ± 0.11 and 4.7 ± 0.13 vs. 3.8 ± 0.19, respectively, *p* < 0.05). There were no statistically significant differences for students in grades 11 and 12 (*p* > 0.05, for all).

### 3.3. Relationship of Body Image Satisfaction, Physical Activity Level, and Academic Performance

There were no statistically significant differences for academic performance, across all years of schooling, between adolescents satisfied with their body image and dissatisfied with their body image (*p* > 0.05 for all) (Table 4).

The proportion of satisfaction with body image according to the practice and type of extracurricular activities for the total sample by gender is shown in Table 5. For the total sample, there was a higher proportion of satisfaction with body image in participants who practiced artistic expression (44.1%) and multisport (25.7%) (*p* = 0.003). When stratified by gender, there was a higher proportion of satisfaction with body image in participants practicing artistic expression (44.1%) and multisport (30.8%) (*p* = 0.038) in the females only. There were no statistically significant differences for males.

Table 6 shows the relationship between the PA level and body satisfaction. In girls, after adjusting for age, better parental education, and body mass index, those with a high level of PA were approximately twice as likely to be satisfied with their body image compared to the girls. Females had a low level of PA (OR = 2.4, *p* = 0.031). There were no significant associations between body satisfaction and physical activity level in the boys.

## 4. Discussion

The aim of the present research was to assess the relationship between the PA levels and body image and academic performance in secondary school adolescents. 

Regarding the relationship between grade point average (GPA) and PA level, as in the study by Deliens et al. [31], no relationship between the GPA and PA levels was found in the present study. These results differ from those published by Fox et al. [32] on English university students where more hours of moderate/intense PA were associated with better GPA. Dyer et al. [33] also concluded that sports participation positively influenced academic performance. Moreover, a meta-analysis suggests that PA, especially school physical education, has benefits on some aspects of academic performance, namely, mathematics and reading [34]. These results are consistent with those presented by Sibley et al. [2], who reported increases in passing specific tests of 25% in writing, 27% in reading, and 31% in mathematics after participating in a 4-year dietary control intervention program in U.S. schoolchildren. 

The fact that our results were not statistically significant did not allow us to affirm the possible benefits of PA practice in terms of academic performance, but according to a systematic review [35], they concluded that while there is evidence suggesting positive associations between PA and academic performance, the latter are inconsistent and the effects of the various dimensions of PA (type, amount, frequency, etc.) on cognitive aspects remain to be explored. 

Fox et al. [32], in a study carried out on English school students, found that practicing more hours of moderate/intense PA was associated with a better grade point average. This may be one of the reasons for why we did not find a relationship between PA and academic performance in the present study. Coupled with this, specifically in grade 10, the average school performance was very high at four out of five, so it was difficult to achieve an increase in academic performance.

All of this makes it necessary to increase physical activity in the school stage. Despite this, and the fact that health organizations seek to increase physical activity during school hours, educational institutions seek to increase the time dedicated to academic subjects. 

Analyzing the academic performance, only in grade 12 was the academic performance lower than those who practiced ECPA (Extra-curricular physical activity), and in grades 10 and 11, academic performance was higher than those who practiced AFEC. Academic performance was not influenced, in this case, by the type of AFEC in grades 11 and 12, however, there was a statistically significant difference in grade 10 in terms of lower academic performance in students who practiced artistic PA compared to students who practiced individual and collective sports, data that are in line with that presented by Kari et al. [36], who stated that, in addition to PA practice, it is important to reflect on the type of practice and sport involvement, as they found evidence of a positive association between participation in competitive team sports and individual competitive sports with higher academic performance. The same conclusions were presented by Fox et al. [32], who mentioned that students from English schools who participated in team sports showed better academic performance as well as Muñoz-Bullon et al. [37] on Spanish university students, referring to the benefits in the academic performance of students who participated in informal sport activities. 

In relation to the above, a longitudinal study conducted by Dyer et al. [33] with American adolescents, on the relationship of school results with sports participation, revealed a significant relationship between PA practice and mathematics subject grade. The same authors also discussed the influence of the parents’ socio-economic status as well as highlighting the significance between “physical activity” and “sports participation”, with the latter attracting factors such as coach supervision for greater school involvement of their athletic children. 

Kari et al. [36] showed the positive relationship between the effects of PA practice, not only on health, but also on the achievement of academic success, highlighting that the practice of PA throughout compulsory education subsequently had an impact on life habits in adulthood. Caram and Lomazi [38] also referred to the importance of habits acquired within the family for adult life, stating that, in a study with Brazilian adolescents, adolescent obesity was associated with parental obesity: mother, odds ratio (OR) 2.86 and father, OR: 2.43 with obesity values before the age of 10 years (OR = 2.26), suggesting early family and community intervention with the aim of preventing and modifying risk behaviors. 

Singh et al. [39] found that continuous participation in sports activities increases student concentration in class. These results suggest that an improvement in academic performance could be produced by increasing the concentration of students. However, more physical activity means that students have less time to study, which can counteract the positive effects that physical activity have on academic performance [31].

Regarding the level of satisfaction with body image, we found different results in both genders. In the boys, we did not find any significant association between satisfaction with body image and the level of physical activity. Girls were twice as likely to be more satisfied with their body image if they were physically active.

As reported by Caram and Lomazi [38] on the prevalence of body image dissatisfaction in women as well as the systematic review by Jimenez-Kimenez-Flores et al. [40], girls were more dissatisfied with their body image compared to boys, with the dissatisfaction levels increasing with the concomitant increase in BMI. Given that the relationships between BMI and physical activity levels are known, we can consider these results to be in line with the results observed in our study. 

Despite using a sample with a slightly older age group than in the present study, Ramos-Jiménez et al. [41] found that about 85% of the participants showed dissatisfaction with their body image, a value similar to that found by us (82.4%). In the same sense, it was found that non-athlete girls, therefore, with a lower PA level than athletes, also showed a higher dissatisfaction with their body image. 

Geographic area seems to influence the results: European studies report that a physically active population has a positive perception of body image, while studies in the Americas report the opposite. This aspect may be one of the reasons for obtaining our results.

Regarding the relationship between body image satisfaction and academic performance, our results did not show positive evidence. Paes et al. [42], in a study with Mexican adolescents, found negative evidence between the association between body image satisfaction and academic performance, stating that students with higher body satisfaction had lower academic performance. 

In contrast, Florín et al. [43], in a study with American adolescents, stated that negative body image perception was positively related to lower academic performance. 

Regarding the limitations of the study, one of the major limitations was the sample recruitment process. Although dozens of secondary schools in the District of Porto were contacted, only nine schools were available to participate in the study. In the future, it is suggested that this study be applied to educational establishments in different districts of the country. 

Another limitation of the study refers to the use of measures self-reported by students including the PA levels, anthropometric measures, and academic performance. In the future, it is suggested that the assessment of PA levels be carried out objectively using accelerometers, that the collection of information regarding anthropometric measurements can be carried out using a scale and a stadiometer, and that the GPA information is collected from the official student register and includes all subjects.

In the future, it is suggested that this study be applied to educational establishments in different country districts. Furthermore, in the future, it is suggested: (i) that the assessment of two levels of PA be conducted objectively using accelerometers; (ii) that the collection of information related to anthropometric measurements could be carried out using a balance and a stadiometer; and (iii) that information about the school media is collected from the official record of the students and includes all disciplines.

## 5. Conclusions

Although there was no association between the PA level and academic performance, it was observed that for the grade 10 students, a higher academic performance was established for students practicing group and individual sports compared to those practicing artistic sports. The social factor, group interaction, and superior coaching, with the attribution of responsibilities and achievement goals, can play an important role in the sporting and academic development of adolescents. The influence of the nature of the sport practiced on GPA and other social and behavioral factors need further study in Portuguese adolescents. 

We found that there was no association between body image satisfaction and academic performance for both sexes, and that girls with higher levels of PA were two times more likely to be satisfied with their body image. Thus, PA, particularly in females, appears to have a decisive contribution to personal satisfaction, self-esteem, and well-being.

## Figures and Tables

**Figure 1 healthcare-11-00602-f001:**
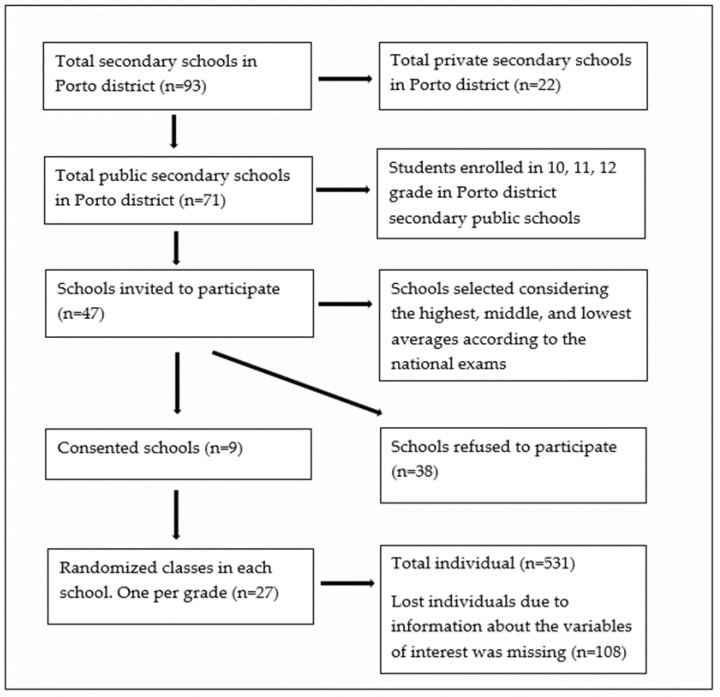
Sample flowchart.

**Figure 2 healthcare-11-00602-f002:**
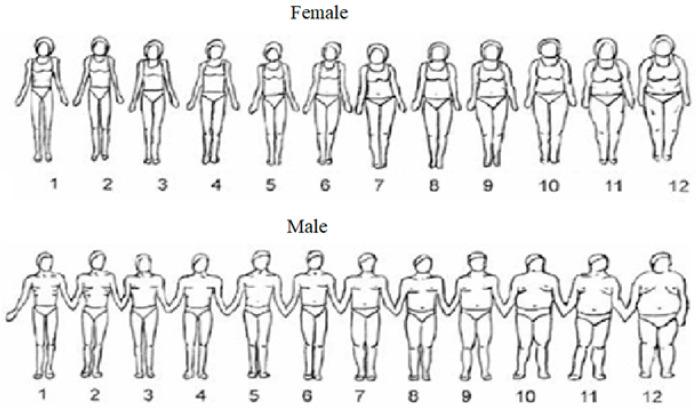
Body Image Rating Scale [27].

**Table 1 healthcare-11-00602-t001:** Characterization of the sample by gender.

	Total (n = 531)	Girls (n = 296)	Boys (n = 235)	*p*
Age, years	17 (16;17)	17 (16;17)	17 (16;17)	0.059
Years of schooling, n (%)				
10th	159 (29.9)	103 (34.8)	56 (23.8)	0.016
11th	172 (32.4)	103 (28.7)	56 (37)	
12th	200 (37.7)	108 (36.5)	92 (39.1)	
Weight, kg	61 (55; 70)	58 (53; 64)	69 (60; 75)	<0.001
Height, m	1.68 (1.62; 1.75)	1.63 (1.59; 1.68)	1.76 (1.71; 1.81)	<0.001
BMI, n (%)				
Normal weight	418 (80.5)	235 (81)	183 (79.9)	0.917
Excess weight	93 (17.9)	51 (17.6)	42 (18.3)	
Obesity	8 (1.5)	4 (1.4)	4 (1.7)	
Academic performance by year of schooling				
10th Grade	4 (4; 5)	4 (4; 5)	4 (4; 5)	0.765
11th Grade	14 (12; 17)	15 (13; 17)	14 (12; 17)	0.354
12th Grade	14 (12; 16)	13 (12; 15)	14 (12; 16.8)	0.209
School motivation	26 (22; 29.3)	26 (22; 30)	25 (22; 28)	0.020
Out-of-school physical activity, n (% of yes)	306 (57.6)	154 (52)	152 (64.7)	0.002
Martial arts	19 (6.2)	6 (3.9)	13 (8.6)	<0.001
Gym sports	47 (15.4)	29 (18.8)	18 (11.8)	
Team sports	106 (34.6)	35 (22.7)	71 (46.7)	
Individual sports	63 (20.6)	36 (23.4)	27 (17.8)	
Artistic expression	34 (11.1)	22.1 (34)	0 (0)	
Multiple	37 (12.1)	14 (9.1)	23 (15.1)	
PAQ-A, points	2.3 (1.8; 2.8)	2.1 (1.7; 2.6)	2.6 (2; 3)	<0.001
Dissatisfaction with body image, n (%)	425 (82.4)	231 (80.2)	194 (85.1)	0.149
Difference between perceived and desired body imagen, n (%)	1 (−1; 2)	1 (−1; 2)	1 (−1; 2)	0.559

BMI, body mass index; PAQ-A, physical activity questionnaire for adolescents; Missing values for the variable’s height (n = 10), weight (n = 7), PAQ-A (n = 81), school motivation (n = 13), body satisfaction (n = 15), average school (n = 35).

**Table 2 healthcare-11-00602-t002:** Comparison of academic performance by year of schooling according to the level of physical activity.

	Physical Activity			F	*p*
	Light	Moderate	Vigorous		
Year of Schooling					
10th	4.4 ± 0.15	4.3 ± 0.12	4.5 ± 0.012	0.529	0.591
11th	14.7 ± 0.35	14.1 ± 0.33	15.1 ± 0.34	1.981	0.142
12th	14.5 ± 0.40	14.4 ± 0.40	13.7 ± 0.49	0.802	0.450

Means ± standard error configured by gender, age, school motivation, and better educational level of the parents.

**Table 3 healthcare-11-00602-t003:** Comparison of the school average, by year of schooling, according to the practice and type of out-of-school physical activities.

	Extracurricular Physical Activity and Practice				Type of Extracurricular Physical Activities							
	Yes (n = 289)	No (n = 207)	F	*p*	Martial Arts (n = 33)	Gym Sports (n = 60)	Team Sports (n = 100)	Individual Sports (n = 18)	Artistic Expression (n = 46)	Multiple	F	*p*
Year of Schooling												
10th	4.4 ± 0.07	4.3 ± 0.10	2.242	0.137	4.5 ± 0.22	4.1 ± 0.24	4.7 ± 0.11 ^a^	4.7 ± 0.13 ^a^	3.8 ± 0.19	3.9 ± 0.27	4.733	0.001
11th	14.8 ± 0.23	14.7 ± 0.30	0.004	0.952	14.8 ± 0.98	15.3 ± 0.62	14.9 ± 0.40	14.9 ± 0.54	14.7 ± 0.70	15.2 ± 0.69	0.116	0.988
12th	14.1 ± 0.25	14.6 ± 0.32	1.552	0.214	13.9 ± 1.07	13.4 ± 0.57	14.2 ± 0.38	13.9 ± 0.58	15.1 ± 0.95	14.1 ± 0.79	0.552	0.736

Means ± standard error adjusted for age, school motivation, and better educational level of the parents; ^a^
*p* < 0.05 compared to artistic expression.

**Table 4 healthcare-11-00602-t004:** Comparison of academic performance for body image satisfaction by year of schooling.

	Satisfaction with Body Image		F	*p*
	Yes	No		
Year of Schooling				
10th	4.4 ± 0.06	4.2 ± 0.15	1.364	0.245
11th	14.6 ± 0.20	14.7 ± 0.39	0.113	0.737
12th	14.1 ± 0.17	13.5 ± 0.48	1.017	0.315

Means ± standard error adjusted for age, gender, school motivation, and the parents’ best educational attainment.

**Table 5 healthcare-11-00602-t005:** Proportion of satisfaction with body image according to the practice and type of extracurricular activities.

	Extracurricular Physical Activity				Type of Extracurricular Physical Activity							
	Yes (n = 294)	No (n = 222)	ꭓ^2^	*p*	Martial Arts (n = 16)	Gym Sports (n = 47)	Team Sports (n = 103)	Individual Sports (n = 61)	Artistic Expression (n = 34)	Multiple (n = 35)	ꭓ^2^	*p*
Sat. with Body Image n (%)												
Total	57 (19.4)	34 (15.3)	1.444	0.229	3 (18.8)	8 (17.0)	16 (15.5)	7 (11.5)	15 (44.1)	9 (25.7)	17.645	0.003
Female	35 (23.8)	22 (15.6)	3.053	0.081	1 (20.0)	6 (20.7)	6 (18.8)	4 (11.1)	15 (44.1)	4 (30.8)	11.795	0.038
Male	22 (15.0)	12 (14.8)	0.001	0.976	2 (18.2)	2 (11.1)	10 (14.1)	3 (12.0)	-	5 (22.7)	1.557	0.816

ꭓ^2^, Chi-square test value.

**Table 6 healthcare-11-00602-t006:** Odds ratio for body satisfaction by level of physical activity and gender.

	Girls (n = 245)		Boys (n = 182)	
	OR (IC 95%)	*p*	OR (IC 95%)	*p*
Model 1				
Under	Ref.		Ref.	
Medium	1.39 (0.61; 3.16)	0.429	1.27 (0.49; 3.32)	0.626
High	2.37 (1.10; 5.13)	0.029	1.16 (0.43; 3.07)	0.773
Model 2				
Under	Ref.		Ref.	
Medium	1.47 (0.64; 3.39)	0.367	1.16 (0.42; 3.26)	0.773
High	2.40 (1.08; 5.32)	0.031	1.22 (0.433; 3.42)	0.709

OR, odds ratio; Ref, reference; Model 1, unadjusted; Model 2, adjusted for age, best parental education, body mass index.

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
