# Peer review of "Physical Activity, Body Image, and Its Relationship with Academic Performance in Adolescents"

_healthcare, 2023, doi:10.3390/healthcare11040602_

Round 1
Reviewer 1 Report (Previous Reviewer 1)
This study aimed to investigate the relationship between physical activity levels and body image and academic performance in secondary school adolescents. The topic is interesting, however, there are some comments.
The manuscript seems to have one part it’s not clear the follow of the abstract structure missing the dots. The sentences are linked together and should be separated.
Thought the manuscript there are many grammatical errors for example
Line 15, ranting should be rating. Please check the text accordingly and correct any grammatical errors.
The introduction should provide a summary background. The current introduction seems too long and provides different information. I suggest to re-write it and making it smaller 4 paragraphs are perfect. No need to write 2 pages!!!
The methods section
The inclusion and exclusion criteria are not clear, please provide clear recruitment criteria.
The authors used different outcome measures and reported them in the methods; however, these outcomes are not reported in the abstract.
The results written presented well.
The discussion section needs improvement as the introduction. The discussion seems too long and the ideas are not connected. The discussion should provide different explanations translating the main results. I suggest organizing the discussion again.
Author Response
Thank you very much for the review. A document with the response to it is attached. Thank you

Reviewer 2 Report (New Reviewer)
Based on the cross-sectional study, this article explored the relationship between physical activity and school performance as well as body image and school performance separately. The purpose of the study was very interesting, but none of the findings seemed to be highly significant and lacked practical significance. The language of the article was complex and lacks logic. Neither the study methodology nor the content of the research seems to address well what the study was trying to achieve.
1. Mathematical and statistical methods were not presented in the abstract.
2. “it was observed that for 10th grade 18 students a higher grade point average was established for students who play sports.” This statement was not supported by clear data in the article and cannot appear in the abstract as a conclusion.
3. The abstract only clarified the findings between physical activity and school performance; however, it did not summarize the findings between body image and school performance.
4. The introduction devoted a significant amount of space to content unrelated to the study's topic, did not highlight the study's key background, a and was cluttered which tended to bore the reader.
5. When exploring the relationship between physical activity level and academic performance in 3.2. Table 2, why was the indicator physical activity level used instead of the indicator physical activity classification (i.e., sedentary, light, moderate, and vigorous activity).
6. Did the authors use the terms “school average” and “academic performance” in 3.2. to mean the same thing? Please standardize the proper nouns.
7. Why did the analysis in Table 3 not adjust for gender?
8. Table 5 needs to be reorganized.
9. I think the main purpose of the article is to explore the relationship between physical activity and academic performance, and the relationship between body image and academic performance. The regression analysis, however, investigated the relationship between body satisfaction and physical activity, which seems to be off the main topic. The dependent variable should be academic performance which can be transformed into categorical variables in the logistic regression model. (Table 5)
10. There are many confusing uses of synonyms in the article (e.g., body satisfaction and body image, physical activity and classification of physical activity, academic performance and school performance, academic achievement and school average, sex and gander, and so on), which to some extent cause confusion to the reader.
Author Response
Thank you very much for the review. A document with the response to it is attached. Thank you

Reviewer 3 Report (New Reviewer)
Dear Authors: I have reviewed the manuscript entitled “Physical activity, body image and its relationship with school performance in adolescents” for consideration to be published in Healthcare. This is an observational study that evaluates physical activity, body image and its relationship with school performance in adolescents. Although potentially useful and interesting, there are several concerns about the methodology that do not make the validity of the study possible.
Materials and methods (page 3 - lines 132-146) :
- Mainly, this reviewer would appreciate a common thread through a flowchart that would clarify the exact sample that was to be chosen, as well as the sample that was finally studied.
- What was the total eligible sample of the nine schools, did they all accept and there were 531? In the event that this was not the case, it should state how many refused to carry out the study, please clarify this issue.
Author Response
Thank you very much for the review. A document with the response to it is attached. Thank you

Round 2
Reviewer 1 Report (Previous Reviewer 1)
I would like to thank the authors for this significant improvement. The manuscript is improved, and the authors followed the comments accordingly. Indeed, there is no more comment to be addressed. I suggest accepting this paper as it is.
Author Response
Thank you very much for the work done and for the comments as this has made the manuscript improve. Thanks.Reviewer 2 Report (New Reviewer)
Most of the comments were answered well, however, comment 9 did not respond and revised.The authors answered that the study aims to establish the relationship between body image and academic performance.Therefore, the dependent variable in logistic regression analysis should be academic performance which can be changed to categorical variables (Table 5). So I suggest adding logical regression analysis,which academic performance looked as reaction variable (Y).
Author Response
Thank you very much for the comments made.Indeed, as the reviewer comments, one of the objectives was to compare academic performance with body image. This aspect is found in Table 4, where satisfaction with body image is compared with academic performance by year of schooling.
Thank you
This manuscript is a resubmission of an earlier submission. The following is a list of the peer review reports and author responses from that submission.
Round 1
Reviewer 1 Report
This study aimed to assess the relationship between physical activity levels, body image, and academic performance in secondary school adolescents. Indeed, different lines of evidence reported a significant and direct relationship between physical activity and academic performance. Confirm these results are still important, especially with the powered sample. I have some major and minor concerns listed below:
Abstract:
-the authors used numbers to categorize the abstract, I feel this is not appropriate, please check the author's instructions and follow the instruction.
-the conclusion is not clear and did not reflect the main results or the main aim of the study.
Introduction
The introduction section is too long, and not organized well. The authors include different ideas without appropriate connections. This section should be improved.
-In line 37, the abbreviation OMS is not defined.
-In lines 46-47 the sentence should be cited.
Line 53 (p.126) is what the authors mean by this.
Lines 59-66 these sentences should be cited.
Materials and Methods
The authors ignored reporting the inclusion and exclusion criteria.
Results
The authors reported the result well and used different statistical analyses to test their hypothesis.
Discussion
The discussion section is not clear, the authors discussed the results generally.
The ideas should be connected depending on the current results.
The authors ignored reporting any limitations related to this study.
The conclusion is general.
Author Response
Thanks for the comments made. All the answers are available in the attached document.
Thank you very much

Reviewer 2 Report
Lines 39, 72, 144 etc - try to use the abbreviation for physical activity, i.e PA.
''But what is PA? PA is defined as "any bodily movement produced by skeletal muscles that results in caloric expenditure" [10] (p.126).'', do we need to define physical activity? I think everyone knows what it means, I don't think it's something new.
''According to WHO [5] in 2010, 81% of adolescents aged 11-17 years considered themselves insufficiently active'', is there no more recent data in this direction? did you look for this information in more recent Eurobarometers?
Please highlight as clearly as possible what was the novelty of this study. Also, I think you should try to shorten the Introduction chapter because it is quite broad and I don't think all the information presented is necessary. Please focus on PA, Body-image and the other variable.
Lines 187-189 - Please include what was the code of the ethics agreement and the date of approval.
From m IPAQ - The authors specify that scores between 8 points to 40 points, please also specify the classifications for these values, not only as "lower scores indicate lower levels of PA'', for example for medium and high values what are the fitting scores?
Figure 1 - I don't think to should remain the names for men and women in other language than English, please change the name of the gender. Also, the quality of the figure must be improved.
Please include in the discussion chapter what were the limits of this study.
Also in this chapter try to correlate more the results obtained with the existing studies, the authors discuss too much about other studies, and less about the results of their own study.
Thank you!
Author Response

(The authors gave the same response as above.)

Reviewer 3 Report
Dear Authors,
You can find attached the document with the review.
Best regards

Author Response

(The authors gave the same response as above.)

Round 2
Reviewer 1 Report
I feel that this manuscript is still need major modification, the methods and results are not clear.
Author Response
A document with the answers is attached.

Reviewer 2 Report
Line 206: Figure 1. Body imagne rating scale, I think image
School average by year of schooling 10º año? also 11 and 12
Author Response

(The authors gave the same response as above.)

Reviewer 3 Report
- The review document is attached.

Author Response

(The authors gave the same response as above.)
